# CIRCA: CAUSAL INTERPRETABLE READABILITY FOR CHINESE ASSESSMENT

## ABSTRACT

Readability assessment is pivotal in education. The prevailing frameworks have limitations: indirect statistical regressions are constrained by correlational paradigms, failing to uncover the causal mechanisms between text features and readability. Meanwhile, although deep learning-based direct methods have succeeded in prediction, they lack interpretability, which hinders the dynamic optimization of text. Grounded in the Chinese context, we propose the CIRCA framework (Causal Interpretable Readability for Chinese Assessment). This framework disentangles spurious associations from genuine causal effects through mathematically principled counterfactual interventions and develops a quantification model using total variation distance. The results show that features insignificant in correlation analyses can exert substantial causal impacts on readability. The determinants vary by grade: in lower grades, topic ambiguity and lexical richness dominate; while in higher grades, semantic noise is more prominent. The correlation coefficient between readability scores computed using the correlation-based formula and the grade in Chinese Textbook Series (2022 Edition) is 0.63, which is notably lower than the 0.73 achieved using CIRCA, thus demonstrating the superiority of the proposed framework.

## 1 INTRODUCTION

As a core pathway for knowledge acquisition, perspective broadening, and literacy enhancement, reading directly influences the cognitive development of students (Amendum et al., 2018). Empirical studies have shown that when the text difficulty aligns with a student's reading ability, the student can independently and efficiently complete reading comprehension tasks, thereby forming a sustained and positive feedback loop in reading (Treptow, 2006). Therefore, readability assessment plays a vital role in supporting students' development.

The state-of-the-art researches on readability assessment follow two technical paradigms, namely *indirect approaches* and *direct approaches*. The former based on feature engineering, which utilize statistical regression models to quantify relationships between explicit text features and readability (Flesch, 1948; Wu et al., 2020; Cheng et al., 2020; Liu et al., 2021; Sung et al., 2013; Wang, 2017). While the latter driven by deep learning, which employ natural language processing to extract latent semantic representations, enabling end-to-end readability prediction (McNamara et al., 2015; Li et al., 2022; Liu et al., 2024; Jiang et al., 2018; Meng et al., 2020; Qiu et al., 2021).

Both indirect and direct approaches present limitations. While constructing explicit readability measures, the indirect approaches operate within a correlation-validation paradigm that merely confirms statistical associations between readability and text features, failing to address the causal nature between them. To investigate this causal essence, it is necessary to verify whether isolated variations in feature variables inevitably induce changes in readability while controlling for confounding factors, which is a critical gap of indirect approaches. This absence of causal linkage may lead to biased identification of key features, thereby compromising readability assessment with systematic errors.

Direct approaches are inherently black-box mechanisms that compromise interpretability, as they typically use neural networks to map latent text embeddings to readability scores. From a practical perspective, the core value of readability assessment lies not only in static difficulty calibration but also in supporting the dynamic optimization process of a text (Collins-Thompson, 2015). Specifically, it should guide the adjustment of texts through interpretable feature attribution, aligning with

the cognitive load of students. However, direct approaches can only output readability scores and fail to trace the specific contributions of linguistic features to readability. This makes it difficult for educators to diagnose the root causes of text flaws and generate actionable revision suggestions. The lack of this attribution severely limits the application of such methods in scenarios that require feedback and adjustment, such as textbook writing and reading recommendations.

Given the limitations of existing approaches, introducing causal analysis into readability assessment research is of theoretical necessity. However, causal analysis in this domain faces significant methodological challenges. Although Randomized Controlled Trials (RCTs) identify causality by fixing outcomes in advance and manipulating variables under control, this design is infeasible for readability assessment, since texts are written directly rather than created by outcome-driven variable control. The core dilemma stems from the irreversible nature of text creation—authors realize their intentions through writing, while textual feature metrics serve only as posterior descriptions of the writing outcome. The uncontrollability of this generative logic renders the process of reverse-engineering texts via predefined feature parameters fundamentally incompatible with the cognitive principles of human writing. As a result, RCTs are difficult to implement in real-world settings, leaving researchers to rely solely on observational data for causal inference.

To address this methodological dilemma, this paper proposes *the Causal Interpretable Readability for Chinese Assessment (CIRCA) framework*. The CIRCA framework constructs a causal graph based on domain knowledge, formalizing the potential causal relationships between text features and readability through a Directed Acyclic Graph (DAG). Then, it systematically identifies the set of backdoor paths from text features to readability, constructing an adjustment set with the backdoor criterion to identify confounding factors. Furthermore, a causal effect estimation algorithm based on counterfactual reasoning is proposed: by quantifying the distributional shift under counterfactual interventions using Total Variation Distance (TVD)(Billingsley, 1995), the confounding effects among feature variables are mathematically decoupled. This method innovatively quantifies the causal effects of text features on readability, successfully distinguishing spurious correlations from true causal relationships. This mathematically driven causal inference not only circumvents the feasibility barriers of traditional experimental designs but, more importantly, establishes a quantifiable causal law between text features and readability, laying a foundation for constructing interpretable readability assessment.

Another innovation of this paper lies in selecting Chinese as the research object, which differs from prior work that primarily focuses on English readability assessment (Flesch, 1948; Crossley et al., 2017). The difference in research objects leads to distinctions in research methods, as Chinese differs significantly from alphabetic languages in aspects such as morphological structure (e.g., words composed of Chinese characters rather than letter combinations), syntactic features (e.g., sentence boundary ambiguity), and pragmatic habits (Flesch, 1948; Guo & Yao, 2021; Heilman et al., 2007). Adapting to the characteristics of Chinese, this paper discards features universally applicable in English but ineffective in Chinese, such as average word length and number of clauses. Subsequently, based on domain knowledge, eight features with strong descriptive capabilities for readability are selected from both macro and micro levels to fully embody the properties of Chinese.

To the best of our knowledge, this is the first study to apply causal inference to Chinese readability assessment by integrating domain expertise with empirical data, thereby providing a more interpretable and theoretically grounded understanding of textual complexity. Through causal analysis, we uncover several novel findings that go beyond traditional correlation-based approaches:

(1)**Revealing Hidden Causal Features**. After controlling for confounding variables, the analysis shows that lexical collocation stability (the degree to which words are easily combined with each other) and lexical response time (the average word recognition latency) exert significant causal effects on readability, despite weak correlations. This challenges conventional feature selection methods and underscores the importance of causal insights over correlation.

(2)**Stage-Specific Readability Drivers**. Causal effects vary across educational stages: for Grades 1–3, topic ambiguity index (reflecting theme overlap or fragmentation), lexical richness, and lexical collocation stability are dominant; for Grades 4–6, semantic noise and lexical richness prevail. These findings stress the need to align complexity assessment with cognitive development.

(3)**Superior Readability Assessment via Causality**. The correlation between readability scores computed using the correlation-based formula and authoritative textbook grading (Ministry of Ed-

ucation of P.R.C, 2017) is 0.63, which is notably lower than the correlation of 0.73 achieved using CIRCA, thus demonstrating the superiority of the proposed framework.

The rest of paper is organized as follows: The proposed framework is detailed in Section 2, encompassing causal graph construction and causal effect metric. Experiments utilizing Chinese elementary textbook data are described in Section 3, followed by the conclusion in Section 4. In addition, We present the related work in Appendix A.

## 2 THE CIRCA FRAMEWORK

The CIRCA framework (Figure 1) implements causal decoupling between text features and readability. Its first step is to construct a qualitative causal graph based on domain knowledge, which captures the topological relationships between text features and readability, and helps reveal potential dynamic interactions. On the second step, we perform counterfactual interventions on text features in the causal graph using graphical independence criteria and Bayesian theorem (José M. Bernardo, 1994), blocking backdoor paths to eliminate confounding, and isolating unconfounded causal relationships from confounded data. Its third step is utilizing total variation distance (TVD) to quantify causal effects between key variables, deriving an interpretable readability formula based on causal effect strength.

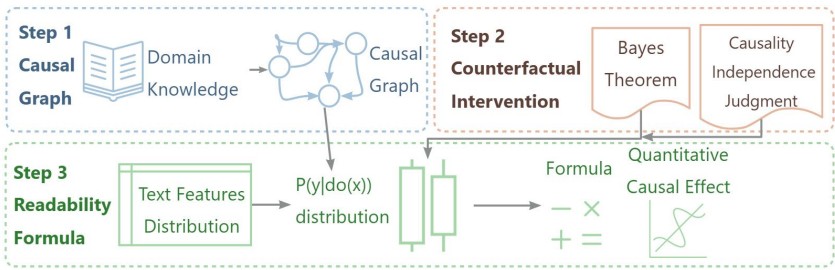

Figure 1: The CIRCA framework.

### 2.1 CAUSAL GRAPH CONSTRUCTION

We systematically select core text features through dual analytical lenses, microscopic linguistic units, and macroscopic semantic architectures to establish a text feature system.

#### 2.1.1 TEXT FEATURE SELECTION

At the micro level, we propose four dimensions:

(1) **Lexical Richness (LR)**, quantified via the entropy of lexical probability distributions to capture vocabulary diversity directly from the text's statistics;

(2) **Syntactic Richness (SR)** is computed as the entropy of dependency relations based on their distributional probabilities in the text (Larson & Ryokai, 2010);

(3) **Lexical Collocation Stability (LCS)**, calculated as the average T-score (Chinese character co-occurrence probability level) from the Chinese Lexical Database (Sun, 2018), to quantify the stability of lexical combinations;

(4)**Lexical Response Time (LRT)**, derived from MELD-SCH psycholinguistic database (Tsang et al., 2018), computing the mean lexical recognition time on words to directly reflect cognitive processing demands.

At the macro level, we employ Latent Dirichlet Allocation (LDA) modeling (Lee et al., 2021) and polysemy analysis to capture semantic dynamics through four dimensions:

(1) **Topic Ambiguity Index (TAI)**, measures how dispersed topics are by weighting topic probabilities according to their rank, to reflect theme overlap or fragmentation (Lee et al., 2021);

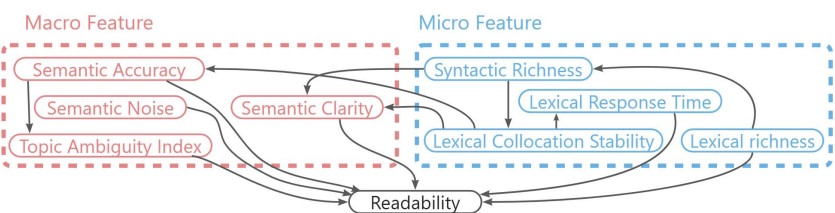

Figure 2: The causal structure of all features.

(2) **Semantic Clarity (SC)**, calculated as the average difference between the highest topic probability and the rest, to reflect how clearly a dominant theme stands out;

(3) **Semantic Noise (SN)**, measured by the kurtosis of the topic probability distribution, to detect redundant or off-topic content through distribution peakedness;

(4) **Semantic Accuracy (SA)**, measured as the average number of senses per content word (McNamara et al., 2015), to reflect internal semantic precision without relying on external difficulty norms.

By integrating micro-level cognitive constraints with macro-level semantic coherence, our framework moves beyond reductionist models and enables a causally interpretable view of readability.

### 2.1.2 CAUSAL PATHWAY ANALYSIS

We construct a causal graph using multidimensional metrics to model readability. As shown in Figure 2, it reveals how micro- and macro-level features jointly influence reading cognition.

Four core features emerged in direct effect mechanisms: Lexical Response Time, Semantic Clarity, Semantic Noise, and Semantic Accuracy. Increased Lexical Response Time indicates accumulated lexical complexity effects, significantly increasing readers' cognitive load. Texts with prominent and concentrated main topics demonstrate higher Semantic Clarity, leading to better comprehensibility. Conversely, elevated Semantic Noise reflects increased redundancy, with such semantic interference substantially distracting readers' attention through irrelevant content, thereby impeding comprehension. Regarding Semantic Accuracy, high-frequency content words with multiple senses show lower accuracy values, whereas simplified lexical choices (higher accuracy) enhanced readability.

Lexical Collocation Stability mediated reading cognition through three distinct pathways. First, collocational rigidity significantly predicted reduced lexical retrieval time. Second, conventional collocations improved thematic coherence by enhancing semantic predictability, which elevated Semantic Clarity scores. Third, increased collocation stability modulated Semantic Accuracy, as evidenced by concurrent rises in both metrics when stable high-frequency words predominated.

Syntactic Richness exerted multidimensional effects through two primary pathways. First, it directly predicted Lexical Collocation Stability and Semantic Clarity. Complex syntactic patterns compelled atypical vocabulary, which disrupted collocational stability. Simultaneously, increased dependency nesting in sentence structures impaired topical salience detection, reducing Semantic Clarity scores.

Finally, Lexical Richness directly affects Readability. More diverse vocabulary corresponds to higher entropy values, indicating more complex and diversified textual expressions. Simultaneously, Lexical Richness influences Syntactic Richness. High Lexical Richness implies more varied vocabulary is included, which leads to diversified expressions and more complex syntactic structures.

### 2.2 COUNTERFACTUAL INTERVENTION

Following causal graph construction, we systematically use counterfactual interventions and causal identification techniques to analyze interaction mechanisms among variables. The critical task lies in differentiating causal paths from non-causal paths: causal paths propagate effects along causal arrow directions, whereas non-causal paths induce spurious correlations through structures with backward-pointing arrows.

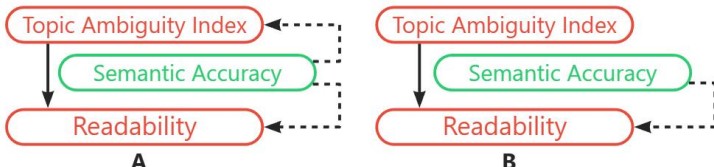

Figure 3: Comparison of pre- and post-intervention.

We focus on non-causal paths with fork structures, where covariate $Z$ serves as a common cause for both $X$ and $Y$. These paths create confounding between spurious associations and actual causal effects, a phenomenon formally defined as the backdoor path. To address this confounding, we introduce a backdoor adjustment set through strategic variable selection to block such paths.

Formally, in a causal graph $G$ with treatment $X$ and outcome $Y$, a backdoor adjustment set $\Phi(X) = \{\phi_1, \phi_2, \ldots, \phi_j\}$ ($|\Phi(X)| = j$) must satisfy:

**Condition 1.** $\Phi(X)$ contains no descendants of $X$.

**Condition 2.** $\Phi(X)$ satisfies the backdoor criterion by blocking non-causal paths between $X$ and $Y$.

Using $X =$ Topic Ambiguity Index and $Y =$ Readability as an example (Figure 3A), path analysis demonstrates that controlling the "Semantic Accuracy" node suffices to block backdoor paths represented by dashed lines. Following this paradigm, we systematically derive complete backdoor adjustment sets between all graph nodes and Readability, as in the third column of Appendix B.

Upon identifying the backdoor adjustment set $\Phi(X)$, causal effects are quantified by eliminating confounding biases. For $Y = y$ under intervention $X = x$, the causal effect is formally defined as $P(y|\text{do}(x))$, where the do-operator implements a surgical intervention that sets $X = x$ while eliminating confounding. Although experimental randomization is infeasible for text features, counterfactual reasoning through mathematical derivation isolates unconfounded causal effects from observational data. This approach rigorously identifies causal mechanisms linking features to readability. First, employing the backdoor adjustment set through the Bayesian theorem yields:

$$P(y|\text{do}(x)) = \sum_{\phi_j \in \Phi(X)} P(y|\text{do}(x), \phi_j) P(\phi_j|\text{do}(x)). \tag{1}$$

As established by Condition 1, the set $\Phi(X)$ contains no descendants of $X$. This structural property ensures the invariance of $P(\phi_j|\text{do}(x))$ under intervention, allowing simplification to:

$$\sum_{\phi_j \in \Phi(X)} P(y|\text{do}(x), \phi_j) P(\phi_j|\text{do}(x)) = \sum_{\phi_j \in \Phi(X)} P(y|\text{do}(x), \phi_j) P(\phi_j). \tag{2}$$

The satisfaction of Condition 2 by $\Phi(X)$ guarantees the blocking of all non-causal paths between $X$ and $Y$. When conditioning on both $x$ and $\phi_j$, the do-operator becomes redundant as the adjustment set $\Phi(X)$ sufficiently accounts for confounding effects. This permits the reduction: $P(y|\text{do}(x), \phi_j) = P(y|x, \phi_j)$. Thus, the causal effect reduces to:

$$P(y|\text{do}(x)) = \sum_{\phi_j \in \Phi(X)} P(y|x, \phi_j) P(\phi_j). \tag{3}$$

This derivation rigorously disentangles spurious associations through causal calculus, enabling precise identification of text features' causal impacts on readability.

Consider the concrete case where $X$ represents the Topic Ambiguity Index, and $Y$ denotes the Readability. When estimating causal effects through conditioning on both $\Phi(X) =$ Semantic Accuracy and $X$ itself, three critical conditions are satisfied: (1) All backdoor paths between $X$ and $Y$ are blocked, (2) No causal pathways between $\Phi(X)$ and $X$ remain active, and (3) The adjustment set $\Phi(X)$ fulfills the Condition 1. This conditioning strategy transforms the original causal graph into the modified structure shown Figure 3B, where confounding factors are eliminated through statistical control. The resulting causal estimand becomes:

$P(\text{Readability}|\text{do}(\text{Topic Ambiguity Index})) =$

$P(\text{Readability}|\text{Topic Ambiguity Index}, \text{Semantic Accuracy}) \cdot P(\text{Semantic Accuracy}). \tag{4}$

Complete derivation matrices for all features are organized in the 4th column of Appendix B.

## 2.3 CAUSAL EFFECT METRIC

Having established the causal identifiability, we quantify the magnitudes of the causal effect using the TVD. This metric operationalizes the discrepancy between interventional distributions and a reference distribution, formally expressed as:

$$\text{TVD}(P(Y|\text{do}(X_i))) = \sum_{x \in X_i} P(\text{do}(x)) \left( \sum_{y \in Y} |P(y|\text{do}(x)) - P(Y = y|U)| \right). \quad (5)$$

The reference distribution $P(Y|U)$ derived from uniform interventions across all treatment conditions:

$$P(Y = y|U) = \sum_{x \in X_i} P(\text{do}(x))P(Y = y|\text{do}(x)). \quad (6)$$

Under the uniform intervention paradigm with discrete treatments, we set $P(\text{do}(x)) = \frac{1}{N_i}$ ($N_i$ represents all possible values that $X_i$ can take). Then, given all the text features $X_i \in \mathcal{X}$, the readability can be defined as:

$$\text{Readability} = \sum_{X_i \in \mathcal{X}} I(X_i) \cdot \text{TVD}(X_i, Y) \cdot X_i. \quad (7)$$

Where the indicator function $I(\cdot)$ encodes causal directionality:

$$I(X_i) = \begin{cases} +1 & \text{if } X_i \text{ positively impacts readability} \\ -1 & \text{if } X_i \text{ negatively impacts readability} \end{cases}. \quad (8)$$

## 3 EXPERIMENTS

In this section, we validate the effectiveness and reliability of the CIRCA framework based on authoritative real data. The aim is to comprehensively evaluate the framework's performance and practical application value in Chinese readability assessment. All experiments are conducted in Python 3.11 on a 3.80 GHz CPU and 16 GB RAM machine. The related experimental code is available at an anonymous repository for review.

### 3.1 DATA SELECTION AND PROCESSING

In readability research, standardized textbooks have been established as the gold-standard benchmark for readability assessment (Guo & Yao, 2021; François & Fairon, 2012; Heilman et al., 2007), with Chinese-language validation evidenced in recent studies (Cheng et al., 2020; Liu et al., 2021). Following this paradigm, we adopted Chinese Textbook Series (2022 Edition) (Organized by the Ministry of Education, 2017) compiled by the Ministry of Education, which strictly adheres to China's Compulsory Education Curriculum Standards (Ministry of Education of P.R.C, 2017). This state-mandated corpus was developed involving 100+ experts, followed by large-scale pedagogical trials in more than 100 schools with 100,000 students. Prior to nationwide implementation, the National Textbook Committee executed rigorous evaluations, ensuring content authority.

Our dataset comprises 443 authentic textbook passages (266,844 characters) spanning Grades 1-6, where grade levels serve as inverse readability indicators (higher grades = lower readability). Through automated text processing, all passages are systematically quantified across the eight predefined linguistic features described in Section 2.1.

### 3.2 COMPARATIVE STUDY OF CORRELATION AND CAUSALITY

In this section, we conduct a quantitative analysis to compare the strength of correlation and causal effects between text features and readability assessment. On the one hand, the analysis of correlation coefficients (Figure 4) revealed three significant positive associations with grade levels, namely

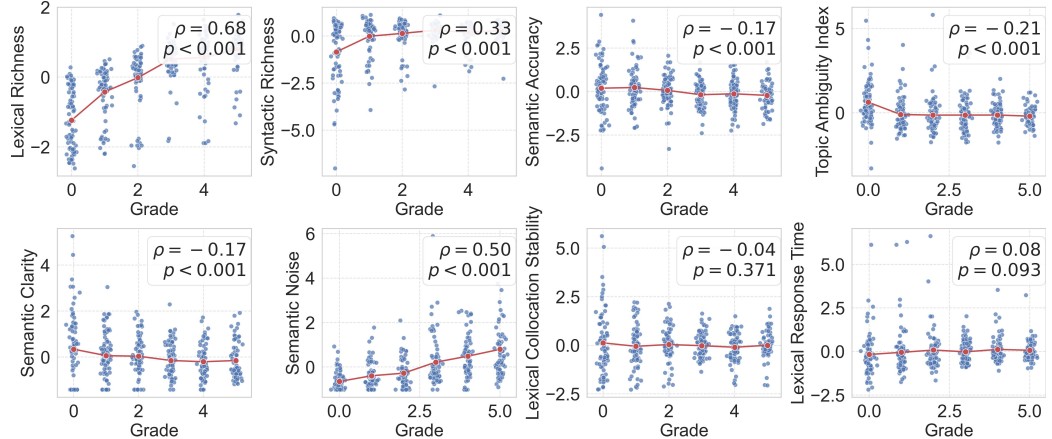

Figure 4: Correlation analysis between textual features and grade-level readability ($\rho$: Spearman correlation coefficient, $p$: Significance level).

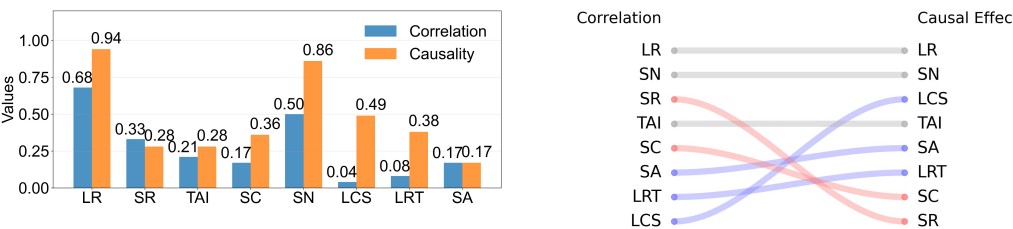

Figure 5: Correlation vs. Causality value.

Figure 6: Correlation vs. Causality rank.

Lexical Richness (LR) ($\rho = 0.68, p < 0.001$), Syntactic Richness (SR) ($\rho = 0.33, p < 0.001$), and Semantic Noise (SN) ($\rho = 0.50, p < 0.001$). This result indicates that the improvement in readability corresponds to systematic linguistic evolution, manifested as: (1) diversified lexical selection; (2) increased structural complexity in syntax; (3) progressive incorporation of complementary thematic elements in semantic space-all contributing to the enhancement.

On the other hand, in Figure 4, significant negative correlations exist for Semantic Accuracy (SA) ($\rho = -0.17, p < 0.001$), Topic Ambiguity Index (TAI) ($\rho = -0.21, p < 0.001$), and Semantic Clarity (SC) ($\rho = -0.17, p < 0.001$). The reduction in Topic Ambiguity Index proves particularly educationally relevant: lower-grade texts employ a breadth-oriented narrative strategy featuring polycentric theme distribution and flexible conceptual boundaries, aligning with students' cognitive abilities; whereas higher-grade texts adopt depth-oriented exposition focused on unitary theme exploration. This progression mirrors cognitive maturation and necessitates advanced information synthesis capabilities in readers.

Notably, in Figure 4, Lexical Collocation Stability (LCS) ($\rho = -0.04, p = 0.371$) and Lexical Response Time (LRT) ($\rho = 0.08, p = 0.093$) showed non-significant associations. Causal path analysis indicates potential confounding effects obscure their theoretical relationship with readability. Given their expected positive correlations per theoretical frameworks, controlled experimental verification is warranted to elucidate their operational mechanisms.

Then we measure the causal effect based on equation 5. The result is shown in Figure 5 and Figure 6. It demonstrates that Lexical Richness (LR) (TVD = 0.94) and Semantic Noise (SN) (TVD = 0.86) exert substantially stronger causal effects on Readability assessment than their correlation coefficients suggest, confirming their dominant roles in determining text complexity. Both causal magnitude and correlational significance consistently identified these as primary determinants.

Notably, after adjusting confounding factors, Lexical Collocation Stability (LCS) (TVD = 0.49) and Lexical Response Time (LRT) (TVD = 0.38) exhibited significantly enhanced causal effects

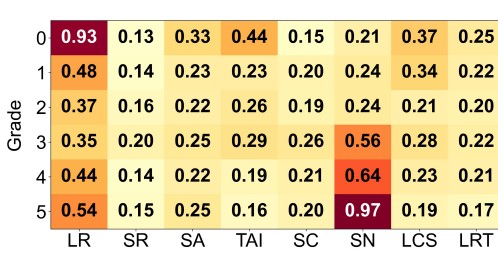

Figure 7: Grade-stratified analysis.

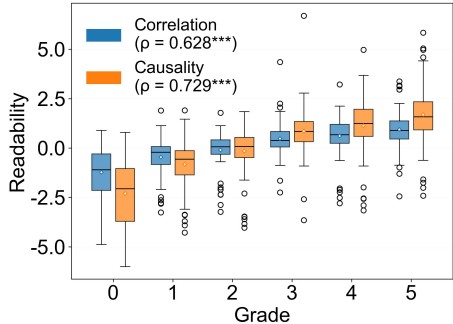

Figure 8: Comparison between correlation and causality.

compared to their original correlations, while improving their feature rankings. The divergence suggests traditional correlation-based approaches systematically undervalue these parameters' functional importance.

Other features show mixed patterns: while Topic Ambiguity Index (TAI) (TVD $= 0.47$) and Semantic Accuracy (SA) (TVD $= 0.45$) demonstrate moderate causal effect enhancements, Semantic Clarity (SC) (TVD $= 0.36$) experiences relative ranking decline despite effect size improvement. Conversely, Syntactic Richness (SR) displays reduced causal influence (TVD $= 0.28$) and ranking descent, indicating its purported role in readability derives primarily from confounding variables rather than direct mechanistic effects.

### 3.3 Cross-Grade Determinants of Text Features

To better analyze the features across grades, we once again applied 5 to measure the causal effects of text in different grades. We re-evaluated causal effects of text features on readability assessment using the CIRCA framework (Figure 1). The analysis reveals different developmental patterns (Figure 7): lower grade texts exhibits the strongest causal impacts of Lexical Richness (LR) (TVD $= 0.93$ at Grade 1) and Topic Ambiguity Index (TAI) (TVD $= 0.44$ at Grade 1), while higher grade texts maintained Lexical Richness (TVD $= 0.54$ at Grade 6) as significant while demonstrating substantially improved causal contributions from Semantic Noise (SN) (TVD $= 0.97$ at Grade 6).

It's worth noting that remaining features display non-stratified causal persistence, with measurable effect magnitudes consistently observed across all educational stages. This pattern suggests grade-invariant mechanisms operate alongside developmentally specific linguistic determinants in shaping readability.

### 3.4 Validation of the CIRCA framework

Next, we validate the proposed CIRCA framework with empirical data. Specifically, we apply (7) for readability assessment across all samples. These results are compared with the corresponding text grade levels, while the Spearman correlation coefficient measures their consistency. The analysis shows $\rho = 0.73$, indicating a significant positive correlation. This verifies that our evaluation method accurately reflects readability and demonstrates strong consistency with the reference dataset.

Moreover, as described in Section 3.2, to comprehensively compare the performance of correlation coefficients and TVD measurement, we replace TVD$(\cdot)$ in equation 7 with the corresponding correlation coefficient for the same calculation, which is called correlation-based formular. The analysis in Figure 8 reveals that the correlation coefficient between readability (via the correlation-based formular) and authoritative textbook grading (Ministry of Education of P.R.C, 2017) is $0.63$, which is lower than the correlation coefficient between readability via equation 7 and authoritative textbook grading ($0.73$). The result strongly supports the superiority of the method proposed in this paper.

In conclusion, equation 7 in the proposed CIRCA framework effectively simulate the distribution characteristics of textbook difficulty in the compulsory education stage. This finding not only ver-

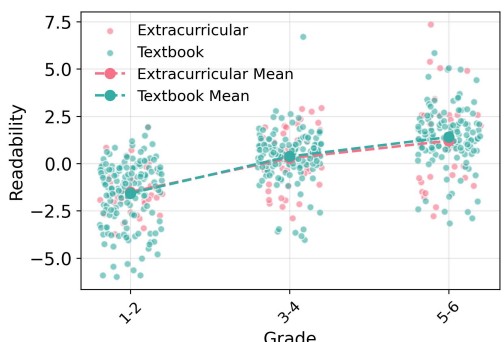

Figure 9: Extracurricular book vs. Textbook.

ifies the rationality of the formula but also suggests its potential application value in the education field, such as assisting with textbook compilation, reading ability assessments, or personalized reading recommendation.

## 3.5 EXTERNAL VALIDATION WITH CURRICULUM-ALIGNED TEXTS

To further validate the generalizability of the proposed readability formula, we conduct an additional experiment using texts from Reading Guidance Catalog for Primary and Secondary School Students (2020 Edition) by the Ministry of Education (Development Center for Basic Education Curriculum and Teaching Materials, Ministry of Education). This catalog provides standardized reading materials aligned with national curriculum objectives. We select 15 books per grade group (1–2, 3–4, 5–6) and extract 117 text passages in total. For each group, we compare readability scores between textbooks and extracurricular books using independent two-sample t-tests.

As shown in Figure 9, the formula-derived readability scores exhibited consistent alignment between textbook and extracurricular texts across all grade levels. For grades 1–2, mean scores are $-1.51(\pm 0.21)$ for extracurricular books and $-1.58(\pm 0.13)$ for textbooks ($p = 0.775$). Similarly, grades 3–4 show comparable scores: $0.28(\pm 0.21)$ for extracurricular books versus $0.37(\pm 0.11)$ for textbooks (p=0.702). In grades 5–6, extracurricular books' scores are $1.20(\pm 0.43)$, while textbooks' scores are $1.40(\pm 0.13), (p = 0.667)$. The absence of statistically significant differences ($p > 0.05$) across all comparisons underscores the CIRCA framework's robustness in evaluating diverse text types while maintaining alignment with curriculum standards. These results confirm that the CIRCA framework generalizes effectively beyond textbook corpora, reliably assessing readability across both instructional and supplementary materials.

In conclusion, CIRCA achieves consistent performance across text genres, reinforcing its practical utility for curriculum design, adaptive learning, and reading assessments.

## 4 CONCLUSION

This paper introduces the CIRCA framework, which integrates causal inference into Chinese readability assessment through counterfactual reasoning and TVD quantification, addressing limitations of prior correlational methods. This approach overcomes correlation-based constraints, uncovering hidden causal determinants like lexical collocation stability and response time despite weak correlations, while providing interpretable, grade-specific insights—from Topic Ambiguity Index and Lexical Richness in lower grades to Semantic Noise in higher ones—aligned with cognitive development theory. Empirical results demonstrate CIRCA's superiority, achieving a 0.73 correlation with authoritative textbook grading compared to 0.63 for correlation-based formulas. Overall, this work offers novel perspectives and methodologies for readability assessment research.

ETHICS STATEMENT

This work does not involve human subjects, animal experiments, or any potentially harmful insights. The datasets used in the experiments are publicly available, and all data processing steps were performed in compliance with the relevant privacy and security regulations. The authors confirm that they have read and adhered to the ICLR Code of Ethics. No conflicts of interest exist in this study, and no financial sponsorship influenced the research.

LLM USAGE STATEMENT

Large Language Models (LLMs) were used in this work for two purposes: (i) language polishing of the manuscript, and (ii) assistance in writing code implementations. The authors take full responsibility for the content and results presented.

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

# APPENDIX

## A  RELATED WORK

Current readability assessment methodologies follow two fundamentally different paradigms: feature engineering-based approaches and deep learning-driven approaches.

The indirect paradigm follows a two-stage framework. Initially, researchers construct linguistically motivated feature sets based on domain knowledge and then apply linear regression linear regression to model feature-readability relationships. For instance, Liu et al. (2021) developed a Chinese textbook readability formula using lexical diversity features; Wang (2017) formulated a learner-specific formula incorporating linguistic features; Wu et al. (2020) identified lexical features as strongly predicting readability in textbooks; Cheng et al. (2020) further advanced feature-based formulas with character frequency and semantic richness.

Advancements in machine learning have prompted scholars to enhance indirect approaches with nonlinear modeling techniques such as Support Vector Machines (SVMs) and Random Forests. For example, Sung et al. (2013) developed Chinese-specific readability metrics implemented via SVMs; Liu et al. (2024) further integrated diverse linguistic features into robust models using Random Forest and SVM.

In parallel, direct methods leverage neural architectures for Chinese readability assessment. For example, Qiu et al. (2021) synergized syntax-aware embeddings with BERT via interaction graphs for holistic evaluation; Li et al. (2022) furthered this by fusing traditional features with BERT embeddings for classification.While these end-to-end models achieve SOTA predictive accuracy, their opacity in decision-making processes—specifically, the difficulty in pinpointing specific linguistic features influencing readability—limits their practice in education.

Current readability assessment methodologies operate under two paradigms: (1) indirect approaches based on feature engineering, which evaluate readability through statistical models linking manually designed features to readability; (2) direct approaches driven by deep learning, where neural networks autonomously extract latent semantic features for end-to-end readability prediction. These paradigms differ fundamentally in methodological orientation.

## B  SUMMARY OF ADJUSTMENT SETS AND DO-OPERATOR CALCULATIONS

| X | Y | $\Phi(X)$ | $P(Y|\text{do}(X))$ |
|---|---|---|---|
| LR | RE | $\emptyset$ | $P(\text{RE}|\text{LR})$ |
| SR | RE | {LR} | $P(\text{RE}|\text{LR, SR}) \cdot P(\text{LR})$ |
| SA | RE | {LCS} | $P(\text{RR}|\text{SA, LCS}) \cdot P(\text{LCS})$ |
| TAI | RE | {SA} | $P(\text{RE}|\text{TAI, SA}) \cdot P(\text{SA})$ |
| SC | RE | {SR, LCS} | $P(\text{RE}|\text{SC, SR}) \cdot P(\text{SR}) + P(\text{RE}|\text{SC, SR}) \cdot P(\text{LCS})$ |
| SN | RE | $\emptyset$ | $P(\text{RE}|\text{SN})$ |
| LCS | RE | {SA} | $P(\text{RE}|\text{LCS, SA}) \cdot P(\text{SA})$ |
| LRT | RE | {LCS} | $P(\text{RE}|\text{LRT, LCS}) \cdot P(\text{LCS})$ |

| | | | |
|---|---|---|---|
| LR | $\rightarrow$ Lexical Richness | SR | $\rightarrow$ Syntactic Richness |
| SA | $\rightarrow$ Semantic Accuracy | TAI | $\rightarrow$ Topic Ambiguity Index |
| SC | $\rightarrow$ Semantic Clarity | SN | $\rightarrow$ Semantic Noise |
| LRT | $\rightarrow$ Lexical Response Time | | |
| LCS | $\rightarrow$ Lexical Collocation Stability | | |

