# OpenReview forum: "CIRCA: Causal Interpretable Readability for Chinese Assessment"
_ICLR.cc/2026/Conference — ICLR 2026 Conference Withdrawn Submission_

### Official Review · Reviewer_4Suk · 2025-10-21

**Soundness:** 2
**Presentation:** 2
**Contribution:** 3
**Rating:** 2
**Confidence:** 4

**Summary:**

The paper investigates work on causal relationships of linguistics variables with respect to readability levels in the context of Chinese readability assessment. The authors motivate the need for the study by citing that recent works on indirect approaches in readability assessment have focused on correlation-validation paradigm that may have a tendency to be shortsighted on statistical associations rather than actual causal nature. To address this, the authors propose the CIRCA framework which aims to model the causal relationships across Chinese-specific linguistic features and readability . The authors then use total variation distance (TVD) to quantify the causal effect of each linguistic variable. The authors use a dataset of 443 seemingly expert-developed Chinese textbooks across 6 grade levels but did not go further into specifying further information about this dataset. The authors conduct an array of experiments centered on comparing correlation results vs. causal relationships from the CIRCA framework. The authors also validate the causal model with an external smaller dataset and show almost identical pattern of causal trends as the grade level categories increase.

**Strengths:**

I appreciate this direction of study that the authors pursued. Causal relationship analysis is very limited in readability assessment research, much less with lower-resource languages like Chinese. Using causal modelling may indeed provide more solid understanding of what linguistic features do affect readability across languages provided proper data preparation and experiment procedures. I particularly like the ablation experiments done to compare current correlation vs causal relationships and found the evaluation on an external dataset to count towards the strength of the causal model.

**Weaknesses:**

While I appreciate the efforts of the paper to explore causal relations as a stronger measure of influence of linguistic features in Chinese readability, there are several major issues with the work that needs to addressed:

The paper contains vague and unexplained terminology. For example, “We systematically select core text features through dual analytical lenses, microscopic linguistic units, and macroscopic semantic architectures to establish a text feature system.” – What do you mean by dual analytical? Microscopic and macroscopic in what context? These are very confusing without proper additional context based on the task.

The paper claims that the study’s novelty is centered around the Chinese language but various parts of the paper seems to lack actual grounding or justification to this language which confuses the reader regarding this intent. In the text feature selection (2.1.1), the authors only list descriptions of features to be extracted but ignore how each feature is extracted with respect to Chinese linguistics. Where are the formulas for these features? For example, with Lexical Richness, there is a common formula for this that originated from English (Type-Token Ratio (TTR) = number of unique words (types) / total number of words (tokens)), is this the same formula used by the authors? Succinct justification and grounding with respect to the complexity of Chinese language and how this deviates from English should be provided for each feature (micro and macro).

Tthere are missing information about how some features were extracted. For example, in the macro-level features, how is the LDA model trained and with what data? Since it’s LDA, how do you evaluate its topic cohesiveness and topic quality?

Basic but essential information about the data is missing. How is the data distributed across the levels? Are they imbalanced? The paper needs descriptive analysis of the characteristics of the dataset per grade level before modelling. Moreover, how will grade level specific counts affect the causal analysis?

The reason why most previous works explore correlations for readability assessment is that they also perform modelling of linguistic features with higher correlations to readability levels as an informed approach for feature selection when training readability assessment models. Hence, my point is that the paper can benefit from an additional exploration of training simple to complex readability assessment models (say using SVM, RandomForest, etc) using the results from highly-causal linguistic features and compare its performance via accuracy of F1score with highly correlated ones. You can even compare the alignment of a trained model with the result on Figure 7.

Upon reading the paper, I’m getting the impression that the authors’ main goal is to outperform the 0.63 correlation score which might limit the overall contribution of the work. Obviously, a solid causal experiment will have the upperhand against correlation if done properly, regardless if it does not beat that score. Is this the main intent of the author? If so, I highly recommend doing the previously suggested model training experiment to support the fact that you may have obtained a better potential predictive and causal measure of readability in Chinese.

**Questions:**

See supporting questions from above.

The causal pathway analysis only provides a figure and some discussion that is observable. Where are the experiment variables and strength of causal connections with respect to each variable?

---

### Official Review · Reviewer_anQx · 2025-11-01

**Soundness:** 2
**Presentation:** 2
**Contribution:** 3
**Rating:** 4
**Confidence:** 2

**Summary:**

The paper proposes a method to evaluate readability of Chinese texts based on causal relations. A set of micro- and macro-features are determined, and connected in DAG. Then causal interventions are performed to determine the causal relationships between each feature and readability. It is found that causal relations differ from correlations in several cases. The method is then applied to a dataset to evaluate readability. The result shops that the determined readability is better aligned with the gold readability, than using correlations.

**Strengths:**

1. The model is based on causal relations between features. This is a novel way to use features to determine readability. The key difference with the existing methods is that the latter are usually based on correlations.

2. The paper shows on a Chinese dataset that the proposed method can better determine readability than correlation-based methods.

**Weaknesses:**

1. While the principle used in the paper is interesting, the implementation heavily relies on the DAG that seems to be manually designed. This raises several questions: the possible causal relations between the features should be known by the experts; one has to do the design again when more features should be incorporated or when it is applied to a different language. This makes the approach hard to scale and to generalize.

2. The method is described at quite high level. Its concrete implementation on the dataset is not described. For example, it is not clear how the probabilities (especially involving do(x)) are determined.

3. The method is tested on a single dataset. If it is tested on multiple datasets, and on several languages, the results would be more convincing.

4. In addition to comparing to correlation-based method, more baseline methods should be included. The related work mentions several studies on Chinese readability assessment. They could be used as baselines.

**Questions:**

see Weakness.

---

### Official Review · Reviewer_8RNd · 2025-11-01

**Soundness:** 3
**Presentation:** 3
**Contribution:** 3
**Rating:** 4
**Confidence:** 3

**Summary:**

The paper introduces CIRCA, a causal inference framework for Chinese readability assessment that combines a linguistically grounded causal graph, counterfactual interventions, and total variation distance metrics to disentangle true causal effects of text features on grade-level readability. Using 443 passages from the national Chinese Textbook Series and external curriculum-aligned materials, the authors quantify micro- and macro-level linguistic features, estimate causal impacts via backdoor-adjusted interventions, and derive a readability formula whose outputs correlate more strongly with grade annotations than traditional correlation-based regressions.

**Strengths:**

The work thoughtfully motivates the need for causal analysis in readability, articulating why correlation-based regressions and neural predictors fall short for actionable feedback. CIRCA’s feature design spans lexical, syntactic, and semantic dimensions with clear educational interpretations, and the causal graph plus counterfactual identification steps are carefully detailed. Empirical studies compare correlation versus causal estimates, provide grade-stratified insights, and validate the resulting formula both on the textbook corpus and an external set of recommended readings, demonstrating practical relevance. The reported improvement in correlation with official grade levels highlights the potential value of the causal formulation for curriculum development.

**Weaknesses:**

Evaluation is confined to a single curated dataset of primary-school Chinese texts, leaving uncertain whether the causal structure or learned coefficients transfer to other genres, regions, or languages. Although the framework aspires to interpretability, most analyses remain aggregate (e.g., TVD bar charts) without qualitative case studies that show how educators might act on counterfactual insights. The comparison baseline relies on substituting correlations into the same formula; stronger baselines such as modern neural readability models or causal discovery methods are absent. Finally, the approach assumes high-quality grade labels and faithful feature extraction, but data noise, alternative DAG specifications, or estimation errors are not explored.

**Questions:**

How sensitive are the reported causal effects and downstream readability estimates to the choice of causal graph? Did the authors test alternative DAGs or perform any robustness checks on backdoor adjustment sets? What variance arises when recalculating TVD-based effects under bootstrapped resamples, and could confidence intervals be reported to quantify estimation stability? Can the authors illustrate specific counterfactual interventions on example passages to demonstrate how practitioners would revise texts using CIRCA’s guidance? Looking forward, how might the framework accommodate richer outcomes, such as comprehension scores, or expand to languages where lexical resources like MELD-SCH are unavailable?

---

### Note · Authors · 2026-05-22

I have read and agree with the venue's withdrawal policy on behalf of myself and my co-authors.

---

### Meta-Review · Area_Chair_5vmG · 2025-12-29

**Summary:**

All three reviewers recognized CIRCA’s novel causal approach to Chinese readability assessment and its empirical gains over correlation-based methods, but consistent critical concerns place the paper marginally below acceptance: limited generalizability from a single primary-school textbook dataset; over-reliance on a manually designed causal DAG without robustness checks; insufficient methodological detail (vague terminology, unclear Chinese-specific feature extraction); weak evaluation lacking strong baselines and practical case studies; and incomplete data transparency (unreported grade-level distribution). The authors did not provide the rebuttal to the reviewers' comments. Thus I would like to recommend rejection.

**Reviewer Concerns:**

The authors did not provide the rebuttal. Thus the concerns raised by the reviewers may not be addressed.

**Reviewer Scores:**

Given that the authors did not provide the rebuttal, the reviewer scores will be likely unchanged.

---

### Decision · Program_Chairs · 2026-01-26

Reject